# A highly accurate platform for clone-specific mutation discovery enables the study of active mutational processes

Eli M Carrami[1,2], Sahand Sharifzadeh[3], Nina C Wietek[1,2], Mara Artibani[1,2], Salma El-Sahhar[1,2], Tatjana Sauka-Spengler[1,4], Christopher Yau[5], Volker Tresp[3,6], Ahmed A Ahmed[1,2]*

[1]Weatherall Institute of Molecular Medicine, University of Oxford, Oxford, United Kingdom; [2]Nuffield Department of Women's & Reproductive Health, University of Oxford, Oxford, United Kingdom; [3]Ludwig Maximilian University of Munich, Munich, Germany; [4]Radcliffe Department of Medicine, University of Oxford, Oxford, United Kingdom; [5]Institute of Cancer and Genomic Sciences, University of Birmingham, Birmingham, United Kingdom; [6]Siemens AG, Corporate Technology, Munich, Germany

**Abstract** Bulk whole genome sequencing (WGS) enables the analysis of tumor evolution but, because of depth limitations, can only identify old mutational events. The discovery of current mutational processes for predicting the tumor's evolutionary trajectory requires dense sequencing of individual clones or single cells. Such studies, however, are inherently problematic because of the discovery of excessive false positive (FP) mutations when sequencing picogram quantities of DNA. Data pooling to increase the confidence in the discovered mutations, moves the discovery back in the past to a common ancestor. Here we report a robust WGS and analysis pipeline (DigiPico/MutLX) that virtually eliminates all F results while retaining an excellent proportion of true positives. Using our method, we identified, for the first time, a hyper-mutation (kataegis) event in a group of ~30 cancer cells from a recurrent ovarian carcinoma. This was unidentifiable from the bulk WGS data. Overall, we propose DigiPico/MutLX method as a powerful framework for the identification of clone-specific variants at an unprecedented accuracy.

*For correspondence:
ahmed.ahmed@wrh.ox.ac.uk

## Introduction

Next generation sequencing has revolutionized our understanding of the genetic evolution of human cells in health and disease (*Turajlic et al., 2019*; *Zhang et al., 2018*). In bulk cancer genome sequencing, inferring the prevalence of variants, the fraction of cells that harbor a variant, enables the computation of the clonal composition of a tumor. In turn, knowledge of the clonal composition enables the construction of evolutionary trees that tell the story of how a particular tumor has evolved over time (*Turajlic et al., 2019*; *Zhang et al., 2018*; *Gerstung et al., 2017*). Analyzing shared mutations within individual clones can be used to deduce mutational processes that may have been operational during the evolution of a tumor. Understanding what mutational processes have taken place within a tumor and what drives them mechanistically, is highly desirable since this could provide opportunities for therapeutic intervention or for predicting the evolutionary trajectory of a tumor. However, the limitation of the depth of sequencing means that only highly prevalent mutations that occurred early in tumorigenesis can be detected using standard bulk whole genome sequencing (WGS) approaches (*Figure 1A*; *Figure 1—figure supplement 1*). Consequently, the ability to model evolutionary events is limited to early events that have been fixed during the tumor evolution and not recent or current processes (*Turajlic et al., 2019*; *Barber et al., 2015*). This limits the

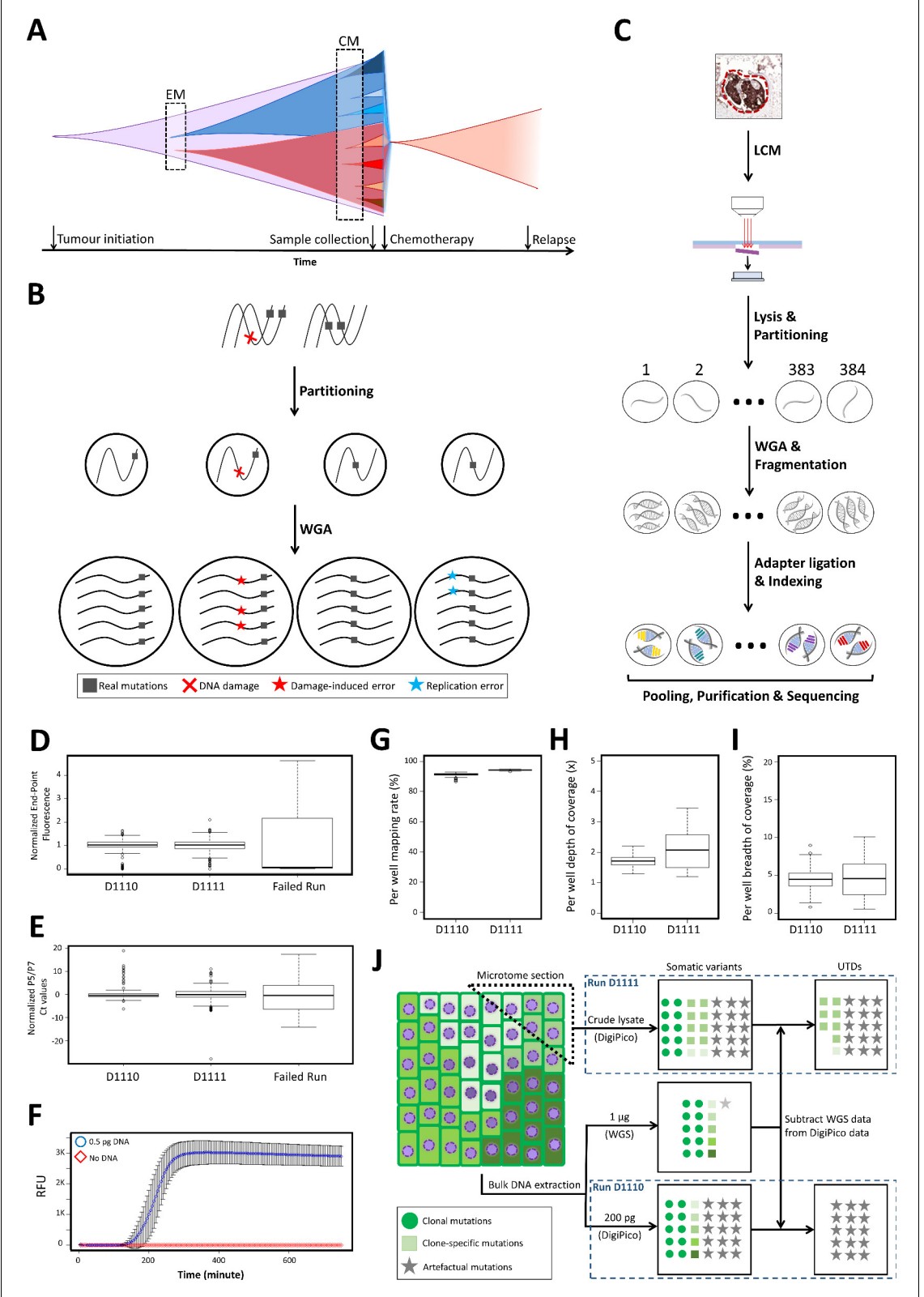

**Figure 1.** DigiPico sequencing rationale, workflow, and performance. (**A**) WGS approaches can only identify early mutational processes (EM) in dominant expanded clones in a tumor (red and blue). Currently active mutational processes (CM) result in a diverse set of sub-clones with different clone-specific mutations. This diversity determines the evolutionary trajectory of the tumor. (**B**) Template partitioning prior to WGA so that each compartment receives no more than one DNA molecule from each locus allows for the identification of artificial mutations. Since damage-induced

*Figure 1 continued on next page*

*Figure 1 continued*

errors (red) and replication errors (cyan) occur stochastically during replication, artefactual mutations result in dual-allelic compartments. (**C**) DigiPico sequencing workflow. LCM: Laser-capture micro-dissection. (**D**) End-point relative fluorescent unit (RFU) from EvaGreen-labeled DNA was used to ensure homogeneous distribution of template and WGA process across the plate. RFU values were normalized to achieve a median of 1 in each run. (**E**) Per well qPCR using Illumina adapter primers (P5 and P7) measures the relative quantity of adapter-ligated products in each well. Ct values were normalized to achieve a median of 0 in each run. (**F**) Streamlining the DigiPico library preparation process required miniaturized a WGA that can specifically and sensitively amplify sub-picogram quantities of DNA in every well. Values represent the mean RFU values across nine replicates. Error bars represent SD. (**G, H, and I**) preliminary analysis of DigiPico sequencing data from individual wells in each run confirms sequencing high-quality and homogeneity of mapping rate, depth of coverage and breadth of coverage as indicated. (**J**) Definition of unique to DigiPico (UTD) variants. Subtracting the SNVs that are identifiable in standard WGS data from corresponding DigiPico data results in UTD variants. These will mainly be consisted of artefactual mutations as well as some clone-specific mutations. Since the template in run D1110 is practically a subset of the template used for the standard WGS, all true variants in the DigiPico run D1110 are expected to also be present in the standard WGS data. In contrast, clone-specific variants in run D1111 are likely to be absent in the standard WGS data because of depth limitation, even though the DNA molecules supporting such variants might have been present in the bulk DNA sample at very low frequencies. In all boxplots the horizontal line represents the median. Boxes represent interquartile range (between the 25th and the 75th percentile). Whiskers represent the range excluding outliers. Outliers are defined as data points above or below 1.5 times the interquartile range.

The online version of this article includes the following source data and figure supplement(s) for figure 1:

**Source data 1.** Raw values for the whole genome amplification monitoring of runs D1110 and D1111.
**Source data 2.** Raw relative fluorescent values for miniaturized whole genome amplification reactions over time.
**Source data 3.** Per well mapping rate, breadth and depth of coverage for runs D1110 and D1111.
**Source data 4.** Sequence of oligonucleotides used in DigiPico library preparation.
**Figure supplement 1.** Challenges in identifying recent mutations.
**Figure supplement 2.** DigiPico library preparation workflow.
**Figure supplement 3.** Analysis workflow for DigiPico data.

practical application of understanding mutational processes. Studying current or recent evolutionary events requires the confident identification of mutations with very low prevalence (*Turajlic et al., 2019*; *Barber et al., 2015*). Sequencing of single cells or small populations of spatially-related cells offer the promise to resolve this issue by the detection of cell-specific or clone-specific mutations (*Figure 1—figure supplement 1*). This gives a readout of mutational processes that are occurring in currently observed cells (*Figure 1A*). However, accurate sequencing of small (picogram) quantities of DNA that can be obtained from single cells or spatially-related cells is highly challenging. Unavoidable DNA damage through oxidation or spontaneous deamination is particularly troublesome when dealing with small quantities of DNA (*Bohrson et al., 2019*). These sources of damage result in disproportionate numbers of artefactual C > A and C > T mutations, respectively (*Bohrson et al., 2019*; *Chen et al., 2017a*; *Costello et al., 2013*). The identification of these artefactual mutations as variants during variant calling results in a large number of false positive (FP) variant calls. An important concern is that such mutations can also be attributed to biological processes such as the accumulation of oxidative DNA damage with age or overactivity of members of the APOBEC family of deaminases (*Nik-Zainal et al., 2012*; *Martincorena et al., 2018*; *Tubbs and Nussenzweig, 2017*). It is not currently possible to differentiate between biologically-driven C > A and C > T mutations and the artefactual ones that have arisen during library preparation when working with picogram quantities of DNA. In addition, the usual step of whole genome amplification (WGA) prior to sequencing inflates the number of artefactual mutations and propagates the errors caused by DNA damage (*Bohrson et al., 2019*). Several approaches have been proposed to reduce DNA-damage during library preparation or to filter out false positive results during analysis (*Bohrson et al., 2019*; *Dong et al., 2017*; *Zafar et al., 2016*; *Chen et al., 2017b*). However, to date, such techniques still result in the retention of thousands of false positive mutations and, therefore, require extensive validation before firm biological conclusions can be made (*Bohrson et al., 2019*; *Dong et al., 2017*; *Zafar et al., 2016*). As extensive validation is not possible in the majority of cases (*Bohrson et al., 2019*), a robust method that eliminates false positive variants from whole genome amplified sequencing data is needed.

In this work we developed a single DNA molecule WGA and sequencing approach to obtain high-quality and data-rich sequencing results from picogram quantities of DNA obtained from clinical samples (we termed DigiPico; for Digital sequencing of Picograms of DNA). Moreover, we implemented a complementary analysis workflow for DigiPico data using an artificial neural network

(ANN)-based algorithm (MutLX, for Mutation Learn) to eliminate FP results while maintaining excellent sensitivity for true positive mutations on a whole genome scale. We validate our approach using data from an extensively sequenced tumor from a single patient with a cumulative depth of ~4200x obtained from 45 WGS runs on DNA from three different time points. We show the versatility of the methods by sequencing samples from four additional cancer patients and a lymphoblastoid cell line.

## Results

### Implementation of DigiPico sequencing approach

A key feature of amplification errors with or without prior DNA damage is that they are introduced at random during the amplification process (*Chen et al., 2017a*; *Costello et al., 2013*; *Arbeithuber et al., 2016*). We, therefore, hypothesized that when amplifying and sequencing a single DNA molecule an artefactual mutation would be present in only a fraction of the reads that have resulted from sequencing the original single DNA molecule. In contrast, genuine variants would be expected to be present in all such reads. Partitioning of the template DNA into individual compartments prior to WGA, such that each compartment receives no more than one DNA molecule from every locus would result in such single DNA molecule sequencing data (*Figure 1B*). Since the artefactual mutations would result in compartments with reads supporting multiple alleles this approach enables the identification of these artefacts and allows for the elimination of FP variant calls. In addition, such partitioning approach also results in the independent progression of WGA reactions for each locus. Thus, providing multiple internal replication data for the WGA process. While genuine variants are expected to be regularly present across replicates, the artefactual mutations, because of their stochastic nature, will likely have a limited presence in a small number of compartments. Consequently, taking both of these points into account, such a WGA and sequencing approach can result in distinctive distribution patterns across the compartments for artefactual mutations compared to real mutations. Since ANNs have shown to be capable of extracting complex patterns from high-dimensional inputs, they make a good candidate for identifying and eliminating FP mutations from this type of data. While previous partitioning and sequencing methods have been described to obtain haplotype information, there is no such method for distinguishing true mutations from artefactual mutations (*Peters et al., 2012*; *Amini et al., 2014*; *Zheng et al., 2016*).

To fully benefit from the data-richness of a partitioning and sequencing approach for accurate genomics study of clinical samples, we developed DigiPico sequencing (*Figure 1C*). To perform DigiPico sequencing, first, we uniformly distribute nearly 200 pg of DNA (obtained from 20 to 30 human cells) into individual wells of a 384-well plate. This ensures that the likelihood for the co-presence of two different DNA molecules from the same locus in the same well is less than 10% (*Peters et al., 2012*). Following WGA, each well is then processed independently into indexed libraries, each receiving a unique barcode sequence, prior to pooling and sequencing (*Figure 1C*; *Figure 1—figure supplement 2*; 'Materials and methods'). Since, in our approach, the key distinguishing factor for artefactual mutations lies in their peculiar distribution pattern, homogeneous distribution and amplification of DNA molecules as well as consistent depth of sequencing coverages across the wells would be critical. Achieving this homogeneity ensures that the differences in the distribution pattern of true and artefactual mutations are maximized. To ensure that the required homogeneity is achieved, during every DigiPico library preparation we monitored the WGA reactions' progress and quantified the final outcome for all the wells. The former was achieved by adding EvaGreen dye to the WGA reactions and monitoring the fluorescent intensity every 5 min in real-time. EvaGreen is an intercalating dye that binds to the minor groove of the DNA and therefore, does not interfere with the isothermal WGA reactions (*Hosokawa et al., 2017*). For the latter, we introduced a per-well qPCR step to measure the relative number of adapter-ligated fragments in each well using adapter specific primers prior to pooling. Only libraries that passed both of these homogeneity tests were used for sequencing (*Figure 1D,E*; *Figure 1—source data 1*). Importantly, we also miniaturized the WGA reaction volumes to 1 μl. This could only be achieved after the identification of a compatible multiple displacement amplification (MDA) approach. Comparing six different MDA strategies, REPLI-g Single Cell amplification was the only method that met the required sensitivity and selectivity for our purpose (*Figure 1F*; *Figure 1—source data 2*). Reaction miniaturization allowed us to streamline the library preparation process in a single 384-well plate without the need for

intermediate purification steps using readily available automated pipetting instruments. Finally, we aimed to optimize the DigiPico library preparation process for frozen clinical samples. This was achieved by performing the WGA reaction directly on the crude lysate of small groups of neighboring cells (tumor islets) isolated via LCM (laser-capture micro-dissection). This strategy ensured the minimal loss of genomic material while minimizing the manipulation time and thereby, reducing the chance of template oxidation.

## DigiPico sequencing platform generates high quality libraries from limited clinical samples

Having optimized all the necessary aspects of DigiPico library preparation process, we decided to assess the quality of DigiPico libraries obtained from clinical samples. For this purpose, we prepared DigiPico libraries D1110 and D1111 from a frozen recurrent tumor sample (PT2R) obtained from a high-grade serous ovarian cancer patient (#11152). In this experiment, while D1110 library was prepared from 200 pg of template taken from a bulk DNA extraction of the PT2R sample, the D1111 library preparation was directly performed on a small frozen section of the remainder of this tumor sample (containing nearly 30 cancer cells). Each library was sequenced on an Illumina NextSeq platform to obtain nearly 400,000,000 reads in $150 \times 2$ paired-end format. The initial assessment of the obtained sequencing data revealed that both the D1110 and D1111 libraries have resulted in high quality sequencing data with an overall mapping rate of 91.35% and 94.27% on human hg19 genome, respectively (*Figure 1G*;; *Figure 1—source data 3*). Analyzing the homogeneity of distribution across the plates indicated that in runs D1110 and D1111, on average, each well covers nearly 4.4% and 4.6% of the genome with an average depth of 1.7x and 2.1x in each well, respectively, with an outstanding homogeneity across the plates (*Figure 1H,I*; *Figure 1—source data 3*). This cumulatively resulted in a breadth of coverage of 92.1% and 91.1% with a depth of 30x and 43x for each run, respectively. These results confirmed that DigiPico sequencing can be used to produce high-quality sequencing data with excellent coverage from limited amount of frozen clinical samples.

Lastly, we assessed whether our initial hypotheses regarding the distinctive distribution pattern of different mutation types hold true in actual DigiPico datasets. For this purpose, we assumed that any variant that is shared between a DigiPico dataset and the standard bulk sequencing data of the same tumor sample must be a true variant. These should mainly consist of germline SNPs and clonal somatic variants. As a result, by definition, all FP variant calls and the majority of clone-specific mutations (had they existed in the sample under study) will be among variants that are only present in the DigiPico data and not in the bulk WGS data. These variants are referred to as UTD for simplicity, hereafter (*Figure 1—figure supplement 3*). Consequently, given that the standard bulk sequencing data of the PT2R sample had been obtained from the same DNA extract that was used for D1110 library preparation, nearly all the UTD variants in D1110 DigiPico run ought to be artefacts (*Figure 1J*). To the contrary, the UTDs in run D1111 are likely to contain some clone-specific mutations alongside the artefactual mutations (*Figure 1J*). Therefore, we used the UTD variants in run D1110 as a representative of artefactual mutations in our analysis. Comparing the frequency of wells with co-presence of two allele for the same locus (*Figure 2A*) as well as the number of wells supporting each variant (*Figure 2B*) in run D1110 showed that UTDs had significantly higher proportion of the former and lower number of the latter compared to any other category of mutations (p<2e-16 for both analyses, One-way ANOVA followed by Tukey HSD testing). This clearly supported the distinct distribution pattern of artefactual mutations in this DigiPico dataset as hypothesized.

## MutLX analysis pipeline for DigiPico data

Having obtained high-quality data using DigiPico sequencing, we decided to implement an analysis pipeline to eliminate FP variant calls based on the distribution pattern of mutations. As mentioned earlier, ANN algorithms are ideally suited for problems with such complex patterns. Given a representative set of correctly labelled examples (training set), an ANN can learn to classify mutations without the need for any class-specific information. However, there are two main issues in implementing ANN algorithms for the problem of eliminating FP mutations from sequencing data; (a) the difficulty in obtaining a generalizable model and (b) unavailability of representative accurately labelled training sets. First, it is not possible to generate a model that is generalizable for the analysis of every DigiPico dataset because the distribution pattern of mutations depends on various run-

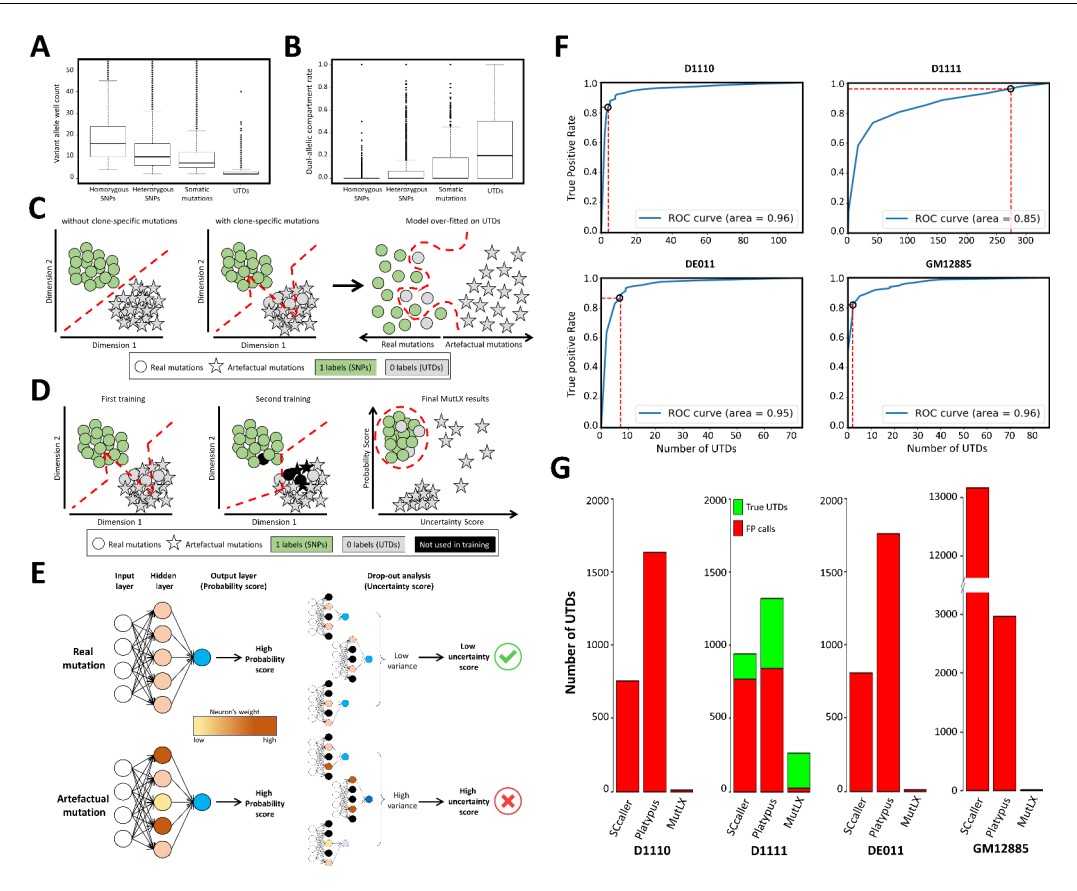

**Figure 2.** MutLX algorithm, design and results. (**A**) Comparing the number of wells supporting various mutation types in run D1110 confirms that, as hypothesized, the majority of UTDs are present in only a few wells. Horizontal lines represent median. Boxes represent interquartile range. Whiskers show the range excluding outliers which are defined as being outside 1.5 times the interquartile range. (**B**) Similarly, the dual-allelic compartment rate of UTDs appears to be significantly higher when compared with true variants. This value was calculated by dividing the number of wells with co-presence of variant and reference alleles to the total number of wells with evidence for variant allele. (**C**) A diagram showing the main challenges in analyzing DigiPico data using ANNs. Each circle/star indicates one variant. Red lines show the behavior of the classification model. All variants above and/or to the left of the lines are predicted to be true variants by the model. The analysis of a sample without clone-specific variant would result in precise separation between real and artefactual mutations. In contrast, the analysis of a sample with true clone-specific mutations would result in a suboptimal model, which could lead to an over-fitting against true UTDs. This will enforce a model that removes all FP calls at the cost of losing nearly all clone-specific variants. (**D**) A diagram showing the two-step training process in MutLX. The first training step identifies some of the mislabeled true mutations (grey circles) among UTDs. All potentially mislabeled data points are temporarily removed from the analysis in the second training (colored in black) so that a better model is obtained for assigning a probability score to all mutations. Finally, combining the probability scores obtained from the model with the uncertainty estimate (as described in E) of these probability scores allows for effective elimination of FP calls while maintaining an excellent sensitivity for true clone-specific variants. (**E**) A diagram showing the test-time drop-out analysis to compute the uncertainty estimate of probability scores. Black neurons indicate the neurons that had been turned off during the drop-out analysis. Accepting only variants with a high probability score and a low uncertainty score should allow for elimination of FP variant calls. (**F**) The ROC curves of the output of MutLX analysis for runs D1110, D1111, DE011, and GM12885 are presented. Circles represent the default cut-off values determined by MutLX. (**G**) Bar plots representing the number of passed UTDs in the output of SCcaller, Platypus and MutLX. Since no true UTDs are expected to be present in runs D1110, DE011, and GM12885 the number of UTDs in these runs represent the FP rate for each analysis method. Values for Platypus are based on DigiPico-specific filtering criteria prior to the application of MutLX.

The online version of this article includes the following source data and figure supplement(s) for figure 2:

**Source data 1.** Comparison of MutLX analysis with Platypus and SCcaller.
**Source data 2.** Targeted sequencing of some of the clone-specific variants identified in run D1111.
**Source data 3.** Targeted sequencing of some of the artefactual variants identified in run DE111.
**Source data 4.** Number of different mutation types in various sequencing runs.
**Source data 5.** Primer pairs used for targeted validation of UTD variants.
**Figure supplement 1.** MutLX analysis algorithm.
**Figure supplement 2.** Analysis of the synthetic DigiPico datasets.

*Figure 2 continued*

**Figure supplement 3.** Frequency of various mutation types among FP calls identified by MutLX in DigiPico data.
**Figure supplement 4.** Probability score values for runs D1110, D1111, DE011, and GM12885.
**Figure supplement 5.** Data simulation confirmed that the AUC negatively correlates with the number of true UTD variants.

specific initial conditions that cannot easily be accounted for (e.g. the copy number state of the genome). Therefore, run-specific models tailored to each DigiPico run will be required. This means that subsets of run-specific mutations need to be selected as a training sets for each DigiPico run. Second, while correctly labelled examples of true mutations can easily be extracted from known SNPs in the genome, identifying a representative and accurate set of examples for artefactual mutations is not possible. To address this issue, we considered UTDs as a reasonable approximation for a representative set of artefactual mutations, assuming that UTDs are predominantly composed of such mutations. This assumption, however, can result in a key challenge. UTDs by definition are composed of artefactual mutations as well as true clone-specific mutations. While artefactual mutations are expected to be abundantly present in all DigiPico runs, true clone-specific mutations may be present at different frequencies depending on the sample (*Figure 1J*). Therefore, when UTDs are all considered as examples for artefactual mutations, samples with more clone-specific variants will have a noisier training set. If this is not taken into account, it can put the samples with true clone-specific variants at an analytical disadvantage, because noisier training sets can result in worse classification models (*Figure 2C*). Specifically, the presence of real mutations among the examples of artefactual mutations in samples with true clone-specific variants may result in over-fitting of the model against these variants (*Figure 2C*). This can undermine the ability of the model to accurately identify true clone-specific variants in such samples. Given that the main objective of DigiPico sequencing is to identify clone-specific variants it is essential to ensure that such over-fitting does not occur when analyzing DigiPico datasets.

Considering all the aforementioned limitations and issues, we designed and implemented an ANN-based binary classifier, MutLX, for the analysis of DigiPico datasets. The focus of the DigiPico analysis pipeline was set for effective elimination of FP calls and accurate identification of true clone-specific variants from UTDs. To address the issue of training with imperfect training sets, we employed the following approach in training MutLX. Initially we considered all UTD variants as examples for artefactual mutations (labelled as 0 s) and a similar number of randomly selected heterozygous germline SNPs as example for true variants (labelled as 1 s). Since the majority of UTD variants are FP calls with an unknown ratio of true clone-specific variants, the 0 labels are considered to be 'noisy' at this stage. In other words, while true clone-specific variants, must have been labelled as 1, because of their anonymity at this stage, they are labelled as 0 among others. To accommodate for this type of noise in the training dataset we employed a two-step training process (*Figure 2D*). In the first step, a model is trained given all labelled example data. This initial model is then used to compute the probability of each mutation belonging to its label's category. Any mutation that appears to have been mislabeled based on these model predictions, is temporarily eliminated (pruned) from the dataset. The underlying assumption here is that even though a model trained on noisy samples might not be as robust as a model trained on a hypothetically clean dataset, still, it will be biased towards better predictions of correct examples because of their higher ratio compared to noise in the dataset. Therefore, the examples that the model predict against their original label, are likely to have been mislabeled in the first place. In the second step, a new classification model is trained on this pruned training set. Since the second training set is likely to contain less mislabeled data points, the final model is expected to be more effective at identifying true mutations regardless of whether or not all UTD variants had indeed been artefacts (*Northcutt et al., 2017*; *Natarajan et al., 2013*). We next employ this model to assign a 'probability score' to each putative mutation. This score indicates the likelihood that a certain mutation belongs to the true variants' category. While this two-step training process is expected to significantly improve the classification model, the final model will still be prone to errors due to imperfections in the training sets. Therefore, we added another level of analysis to further improve the accuracy of our pipeline. This was achieved by assigning an uncertainty estimate to the 'probability score' of each mutation. This uncertainty estimation is based on the assumption that a robust prediction is supported by most of the

activated neurons in the hidden layers of the ANN. Therefore, any subset of these neurons would also consistently result in a similar probability score and hence, there will be a low variance between various 'probability scores' obtained from different neuronal subsets (*Figure 2E*). In contrast, the seemingly high 'probability score' of an artefactual mutation is most likely only supported by some of the neurons in the hidden layer of the ANN. Therefore, different neuronal subsets will result in varying 'probability scores' which will lead to a high variance in the scores obtained from different neuronal subsets for an artefactual mutation (*Figure 2E*). As a result, an 'uncertainty score' can be calculated as the variance of 'probability scores' obtained from multiple different randomly selected neuronal subsets of the trained ANN in MutLX (*Gal and Ghahramani, 2015*). The combination of 'probability score' and 'uncertainty score' for each mutation should therefore allow us to accurately determine whether a called variant is a real mutation or a result of artefactual changes in the template (*Figure 2—figure supplement 1*).

## Validation of the MutLX algorithm

To validate our strategy, we chose to test the MutLX analysis pipeline on runs D1110 and D1111. This is because these DigiPico runs had been obtained from a HGSOC (High-grade Serous Ovarian Cancer) that was previously extensively sequenced with data available from 48 independent WGS data sets across three different time points (patient #11152) at a total depth of approximately 4200x from two independent sequencing platforms (*Hellner et al., 2016*). To our knowledge, this comprises the most extensively whole genome sequenced tumor to date. This exceptionally large dataset allows for reliable cross-validation of mutations in this tumor. For this purpose, we used the MutLX algorithm to analyze the sequencing data from runs D1110 and D1111. As explained previously, when using the bulk sequencing data from the PT2R site for comparison with these DigiPico datasets, true UTD variants (clone-specific variants) are only expected to be present in run D1111, while nearly all UTDs in run D1110 are expected to be artefacts (*Figure 1J*). In addition, we also analyzed DigiPico sequencing data prepared from purified DNA of a blood sample (run DE011), as well as cultured GM12885 lymphoblastoid cells, both of which are also expected to have no true UTD mutations. The de novo variant calling on these DigiPico runs followed by an initial filtering based on the well counts resulted in the identification of thousands of UTD variants in each sample, nearly all of which were expected to be FP calls. However, the application of MutLX algorithm on the UTD variants of runs D1110, DE011, and GM12885 resulted in the effective elimination of over 99% of the FP variant calls in these runs to only 4, 7, and three genome-wide FP mutations, respectively, while maintaining a sensitivity of ~85% for detecting true mutations (*Figure 2F,G*). In comparison, SCcaller (*Dong et al., 2017*) analysis of the same data resulted in 713, 712, and 13,280 FP variant calls, respectively (*Figure 2G*; *Figure 2—source data 1*). On the other hand, MutLX identified 266 putative clone-specific variants in run D1111, 240 of which (90%) were validated through comparison with independent high-depth datasets of this tumor sample (*Figure 2G*). Moreover, these observations were further validated by performing targeted sequencing on the bulk DNA of the tumor. Whereby, out of the 11 analyzed amplicons harboring clone-specific variants from run D1111, 10 were found to conclusively be present at low frequencies in the bulk DNA of the PT2R sample (*Figure 2—source data 2*). Furthermore, amplicon sequencing of 37 seemingly high-quality UTD variants from run DE111 that were labelled as artefactual by MutLX algorithm indicated no evidence for their presence in the bulk DNA sample (*Figure 2—source data 3*). These results clearly confirm that MutLX can learn accurate classification models that distinguish artefactual mutations from real variants and is able to effectively identify true clone-specific variants in DigiPico data.

Additionally, we investigated whether the presence of true clone-specific mutations could compromise the sensitivity of the model due to over-fitting. For this purpose, we artificially mislabeled varying numbers of somatic mutations in runs D1110 and DE111 as artificial UTD variants (UTD*) to generate synthetic datasets with various ratios of true UTDs. These synthetic datasets were then independently analyzed by MutLX and the FP rate as well as the recovery rate of UTD*s at varying UTD*/UTD ratios were examined in all the synthetic datasets. The results showed that a UTD*/UTD ratio as high as 10% does not significantly affect the recovery rate of UTD* variants, indicating that overfitting does not occur in MutLX (*Figure 2—figure supplement 2*).

## Versatility of DigiPico/MutLX sequencing and analysis approach

Finally, to ensure the versatility of our proposed method, DigiPico sequencing was performed on various sources of template DNA from four different HGSOC patients and the resulting UTDs were analyzed using MutLX algorithm. The results clearly indicate that MutLX can reliably identify and eliminate the artefactual variant calls from a diverse set of DigiPico libraries (*Table 1*). This strongly suggests that DigiPico/MutLX can effectively enable the study of recently acquired mutations in solid tumors. Importantly, analyzing the frequency of different mutation types in these data indicated the presence of a higher level of C > A mutations among the identified artefactual mutations, consistent with the notion that such FP calls are a result of oxidative damage to the template DNA (*Figure 2—figure supplement 3*; *Figure 2—source data 4*).

We next tested the feasibility of studying mutational processes in a patient with HGSOC (#11152). For this patient, various sequencing data from a pre-chemotherapy omental mass were available (standard bulk sequencing at 30x as well as five tumor islet DigiPico runs). The patient subsequently had a recurrence and tumor samples were collected from the pelvis (pelvic recurrent tumor; PT2R) and from the para-aortic lymph node (PALNR) for standard bulk sequencing as well as DigiPico sequencing of tumor islets. The analysis of the bulk pre-chemotherapy sequencing data identified 13,721 somatic mutations. 84.6% of these mutations were present in at least three tumor islets from DigiPico data, 91.4% of which were also present in at least three additional islets from previously published LFR data (*Hellner et al., 2016*). The high occurrence of the mutations indicates that they were early mutations that became fixed in the tumor. The analysis of DigiPico data from tumor islets revealed that there was a limited number of clone-specific mutations that were absent in the bulk tumor. Each of the five pre-chemotherapy islets harbored a number of truly unique mutations (2, 6, 8, 8, and 36), compared to other islets, indicating that they were recent occurrences (*Figure 3A*). The bulk WGS data of the PT2R recurrence indicated the emergence of 3009 new somatic mutations that were absent in the pre-chemotherapy bulk sequencing data, DigiPico data or LFR data. These mutations may have occurred at any point since the common ancestors of the omental mass and the PT2R recurrence diverged from each other (*Figure 3A*). The analysis of tumor islets in the recurrence samples from patient #11152 showed that the pelvic recurrent tumor (PT2R) has a high load of clone-specific mutations compared to the para-aortic lymph node recurrence

**Table 1.** Application of DigiPico sequencing and MutLX analysis to a diverse set of clinical samples.

| Run ID | Patient ID | Sample type | Collection site | Sequencing platform | Total UTD | Passed UTDs | Validation rate* | AUC |
|---|---|---|---|---|---|---|---|---|
| D1110 | #11152 | Recurrence | PT2R | NextSeq | 1634 | 4[b] | NA | 0.95 |
| D1111 | #11152 | Recurrence | PT2R | NextSeq | 1325 | 266 | 240/266[c] | 0.85 |
| D1112[*] | #11152 | Recurrence | PT2R | HiSeq 4000 | 1219 | 210 | 189/210[c] | 0.85 |
| D1511 | #11152 | Recurrence | PALNR | HiSeq X | 1786 | 9 | NA | 0.94 |
| D1210 | #11152 | Pre-chemo | OM | NextSeq | 3139 | 28 | 16/28[c] | 0.94 |
| D1211 | #11152 | Pre-chemo | OM | HiSeq 4000 | 5521 | 69 | 17/69[c] | 0.91 |
| D1212 | #11152 | Pre-chemo | OM | HiSeq 4000 | 5015 | 24 | 16/24[c] | 0.94 |
| D1213 | #11152 | Pre-chemo | OM | HiSeq 4000 | 5090 | 46 | 25/46[c] | 0.93 |
| D1214 | #11152 | Pre-chemo | OM | HiSeq 4000 | 3415 | 37 | 27/37[c] | 0.93 |
| DE011 | #11513 | Normal | Blood | HiSeq X | 1759 | 7[b] | NA | 0.95 |
| DE111 | #11513 | Pre-chemo | Ascites | HiSeq X | 3685 | 4[b] | NA | 0.97 |
| D6311 | OP1036 | Pre-chemo | RPCG | HiSeq X | 3185 | 12 | NA | 0.96 |
| DA111 | #11502 | Pre-chemo | LPrt | NextSeq | 12511 | 10 | NA | 0.97 |
| GM12885 | - | Cell line | - | NextSeq | 2970 | 3[b] | NA | 0.96 |

* Run D1112 is a technical replicate of run D1111. [b] Runs where true UTDs are not expected to be present. [c] Validation through comparison with independent high-depth WGS data from the bulk of the tumor. * Validation rate is an under-estimation for the positive predictive value of clone-specific variants. PT2R: Pelvic Tumor Recurrence; PALNR: Para-Aortic Lymph Node Recurrence; OM: Omental Mass; RPCG: Right Paracolic Gutter; LPrt: Left Peritoneum; NA: Not Available; AUC: Area Under the Curve of receiver operating characteristic plot.

The study of active mutational processes using DigiPico/MutLX.

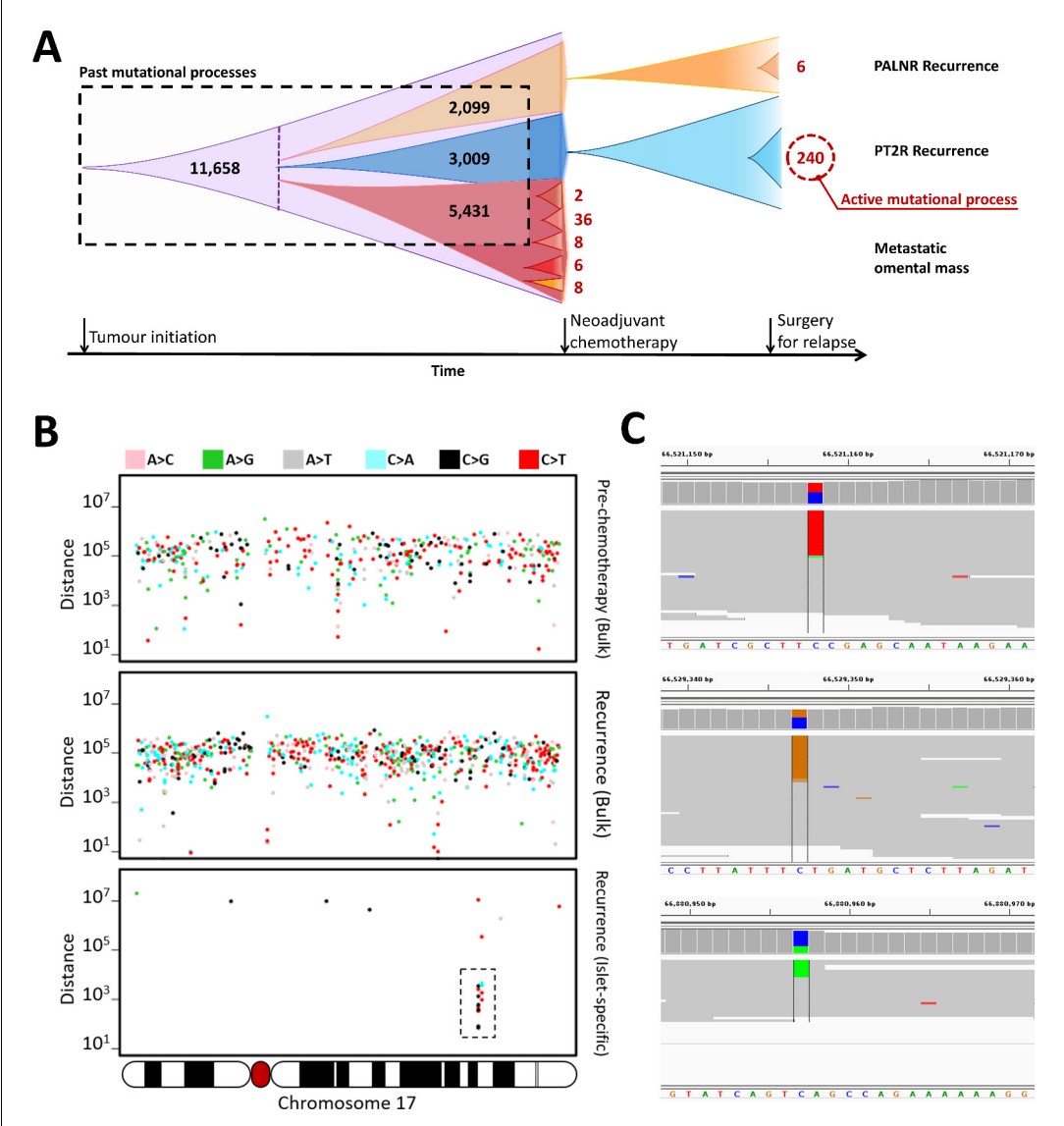

**Figure 3.** Identification of an active mutational process using DigiPico/MutLX. (**A**) Schematic representation of tumor evolution in HGSOC patient #11152. Standard bulk WGS of various tumor samples identified ~11000 shared somatic mutations among all sites. Dotted purple line indicates the point at which the most recent common ancestor of the studied tumor samples has diverged. Bulk sequencing also identified nearly 5000, 3000, and 2000 sub-clonal mutations specific to the pre-chemotherapy omental mass, PT2R recurrence, and PALNR recurrence, respectively. These mutations however could have occurred anytime during the expansion of these clones and is biased towards older mutations. This is due to the limitations in identifying low-prevalence somatic mutations. DigiPico sequencing of five pre-chemotherapy tumor islets, PT2R and PALNR recurrence sites, however, identified various numbers of recently emerged clone-specific mutations in each of these samples (represented in red numbers). The significantly higher number of clone-specific variants in PT2R indicates the presence of an active mutational process. (**B**) This active mutational process is highlighted by the presence of a strong clone-specific kataegis event on chromosome 17 in run D1111. Y-axis represents the pair-wise distance of consecutive somatic mutations in log scale. Only mutations from chromosome 17 are shown. Mutations involved with the sub-clonal Kataegis event are highlighted in the box, nearly all of which are in the form of strand-specific C > T or C > G mutations. This suggests the involvement of APOBEC enzymes in this hyper-mutagenesis process. (**C**) Representative examples of some of the mutations involved in the kataegis. The presence of all the mutations on the forward strand of the genome further confirms the involvement of a hyper-mutagenesis event (*Figure 3—figure supplement 1*).

The online version of this article includes the following figure supplement(s) for figure 3:

**Figure supplement 1.** IGV images of SNVs that were identified in the sub-clonal kataegis in PT2R sample.

(PALNR) or the pre-chemotherapy tumor. This observation suggests that, in this patient, molecular mechanisms underlying the SNV mutagenesis may have been recently activated (*Figure 3A*). Moreover, analyzing the clone-specific mutations of the PT2R sample with a rainfall plot revealed the presence of a strong sub-clonal local hyper-mutagenesis (kataegis) event (*Nik-Zainal et al., 2012*) on chromosome 17 (*Figure 3B,C*; *Figure 3—figure supplement 1*). Comparing the mutations that made up this kataegis events to bulk sequencing data, DigiPico data and LFR data of the pre-chemotherapy omental mass revealed that they were only found in the DigiPico PT2R data indicating that they were genuine clone-specific mutations.

## Discussion

In this work we presented DigiPico/MutLX as an integrated platform for the identification of mutations from small groups of cells with unprecedented accuracy on a whole genome scale using an ANN. While ANNs and other machine learning approaches have frequently been employed in the field of genomics, however, the focus of these methodologies has mainly been on the elimination of technical and computational noise from sequencing data (*Cibulskis et al., 2013*; *Spinella et al., 2016*; *Wang et al., 2013*; *Fang et al., 2015*; *Wood et al., 2018*). Such sources of noise, resulting from low-quality sequencing data or inaccurate mapping of short reads, lead to variant calls that had not actually been present in the sequenced library and their elimination is essential for accurate identification of somatic mutations. However, these methods are unable to address the issue of artefactual mutations that physically occur in the sequenced libraries due to the WGA of damaged DNA. As such, artefactual mutations will exhibit all the features of real mutations and will not be eliminated by the previously described algorithms. Given that the process of WGA 'fixates' such mutations into the final library, it is essential to preserve some information prior to the start of the WGA which can later be employed for identification of these sources of error. In our method, we preserve this information by compartmentalizing the template DNA prior to the start of WGA and by subsequent indexing of the compartments during the DigiPico library preparation pipeline. As a result, in MutLX rather than trying to evaluate whether a variant call had indeed been present in the library, we use the complex pre-WGA information to decipher whether the called mutations had existed in the original template or they have artefactually arisen through the process of WGA. This critical distinction is what allows us to identify extremely low-prevalence mutations in small populations of cells which in turn enables the study of active mutational processes. We, therefore, believe that this work provides an important stepping stone for the discovery of current or recent somatic mutational processes that occur in cancer and normal tissue. It is also important to note that MutLX takes advantage of compartment level data generated through the DigiPico library preparation method followed by relatively deep sequencing of each compartment. However, in theory, other linked-read library preparation methods such as the commercially available 10x platform may also generate data that could be utilized by MutLX, if modifications are applied. For such alternative approaches to benefit from MutLX the DNA input should be lowered, and the depth of sequencing per each compartment must significantly increase.

Understanding current mutational processes is key for predicting the evolutionary trajectory of a tumor and, potentially, for interfering with such trajectories therapeutically. A mutation that is identified in bulk sequencing of a tumor must have occurred at a point during the extended history of a tumor from the initiation till presentation. In contrast, a cell-specific mutation must have occurred during the limited lifetime of that cell. Similarly, a mutation in a small clone that has been derived from a single cell is also recent. The age of such a mutation can't be more than the age of the clone which is defined by the number of cell divisions it took to generate that clone. Studying patterns in cell-specific or small-clone-specific mutations can allow for the identification of recent or current mutational processes (*Turajlic et al., 2019*). Defining such processes is highly desirable since they can be causally linked to biological or chemical phenomena and, therefore, yield significant mechanistic insights. Identifying these mechanisms have important practical implications since they are potentially amenable for therapeutic intervention or for predicting future tumor behavior. The current state of the art does not allow the direct accurate identification of mutations from individual cells or individual small clones from tumors. DigiPico/MutLX enables this endeavor for the first time.

To overcome significant technical pitfalls predominantly related to the discovery of FP mutations, current methods for single cell WGS analysis either require extensive validation studies (*Dong et al.,*

*2017*) or rely on combining data from multiple cells to obtain reliable mutations that are shared between cells (*Zafar et al., 2016*; *Laks et al., 2018*). These cells are then grouped into clones that have been derived from a common ancestor. While such techniques go to a more recent common ancestor compared to bulk sequencing, they are still not ideal as data derived from these approaches do not reflect mutational processes that are taking place in existing cells. Furthermore, reducing the depth of sequencing per cell to enable sequencing large numbers of cells, reduces the breadth of coverage, which is already compromised by loss of genetic materials during preparation steps. This increases the number of cells that are needed to be analyzed for inferring and identifying clones which in turn moves the ancestor further back into the past. In addition, the lack of information about physical relatedness in single cell analysis methods, results in loss of an opportunity to group cells that are likely to be from a single clone. This increases the gap between the ancestor of an inferred clone and the present time, making it difficult to define processes that are active within currently existing cells in a tumor.

DigiPico/MutLX has the distinct advantage of enabling the preservation of spatial information. Analyzing spatially-related cells, preserves physical relatedness and enables the assumption that physically related cells belong to an individual clone (*Martincorena et al., 2018*). Defining distinct structures that may have arisen from a tissue resident stem cell has also been suggested to identify and analyze clones. For example, cells from a single small intestinal crypt or a single endometrial gland could be reasonable expected to come from a single tissue-resident stem cell (*Moore et al., 2018*; *Lee-Six et al., 2019*). Under these circumstances, each anatomical unit defines a clone that may or may not have clone-specific mutations that can be related to a mutational driver. Furthermore, sequencing data from a clone can be computationally used to infer subclones and predict more recent events that may have arisen within a clone. This is akin to what bulk sequencing and analysis achieves but at the level of a single clone that is composed of a limited number of cells. Preserving spatial information is also particularly interesting because of the recent developments in enabling spatial transcriptomics technologies (*Burgess, 2019*). It is conceivable that combining highly accurate DNA sequencing with spatial transcriptomics would allow the dissection of genetic and non-genetic heterogeneity in tissues. In short, current technologies, for the analysis of small clones yield large number of FP results making it impossible to obtain direct accurate clone-specific information on a genome scale without exhausting validation. Combing data from multiple clones, is a common solution but moves the ancestor further back into the past. We have previously used this approach for the analysis of small collection of tumor cells (tumor islets) (*Hellner et al., 2016*). Because of the uncertainty associated with the mutation calls from individual islets, it was necessary to only call mutations that were shared between all tumor islets and effectively identify only truncal mutations. This was then followed by independent validation of some 700 mutations using targeted sequencing. While this still yielded important biological insights, we were unable to study islet-specific mutations. DigiPico/MutLX is now enabling the study of such mutations. We demonstrated how the direct analysis of DNA from ~30 cancer cells, resulted in the confident identification of a subclonal kataegis event.

Overall, here we showed that DigiPico and MutLX can enable hyper-accurate identification of somatic mutations from limiting numbers of cells obtained from clinical samples, as an important improvement over the existing methodologies. Moreover, unlike other computational methods that rely on diploid regions of the genome to calculate amplification biases, our method is also compatible with genomes that suffer from extensive copy number alterations, such as in HGSOC. We believe that the versatility of the DigiPico/MutLX method enables the study of active mutational processes in tumors as well as in normal tissues.

## Materials and methods

### Patient samples and consent

Patients #11152, #11502 and #11513 provided written consent for participation in the prospective biomarker validation study Gynaecological Oncology Targeted Therapy Study 01 (GO-Target-01) under research ethics approval number 11/SC/0014. Patient OP1036 participated in the prospective Oxford Ovarian Cancer Predict Chemotherapy Response Trial (OXO-PCR-01), under research ethics approval number 12/SC/0404. Necessary informed consents from study participants were obtained

as appropriate. Blood samples were obtained on the day of surgery. Tumor samples were biopsied during laparoscopy or debulking surgery and were immediately frozen on dry ice. All samples were stored in clearly labelled cryovials in −80°C freezers (*Table 2*).

## Cell lines

GM12885 lymphoblastoid cell line (RRID:CVCL_5F01) was obtained from Coriell institute and cells were kept in culture as recommended by the provider and were immediately used for DNA extraction.

## Sectioning and LCM

Frozen tumor samples were embedded in OCT (NEG-50, Richard-Allan Scientific) and 10–15 μm sections were taken using MB DynaSharp microtome blades (ThermoFisher Scientific, USA) in a CryoStar cryostat microtome (ThermoFisher Scientific, USA). Tumor sections were then transferred to PEN membrane glass slides (Zeiss, Germany) and were immediately stained on ice (2 min in 70% ethanol, 2 min in 1% Cresyl violet (Sigma-Aldrich, USA) in 50% ethanol, followed by rinse in 100% ethanol. A PALM Laser Microdissection System (Zeiss, Germany) was used to catapult individual tumor islets into a 200 μl opaque AdhesiveCap (Zeiss, Germany).

## Standard WGS and data analysis

DNA was extracted using DNeasy blood and tissue kit (Qiagen, USA). Up to 1 μg DNA was diluted in 50 μl of water for fragmentation using a Covaris S220 focused-ultrasonicator instrument to achieve 250–300 bp fragments. The resulting DNA fragments were then used for library preparation using NEBNext Ultra II library preparation kit (NEB, USA), following the manufacturer's protocol. The resulting libraries were sequenced on Illumina NextSeq or HiSeq platforms at a depth of 30 – 40x over human genome. Sequencing reads in the FastQ format were initially trimmed using TrimGalore (*Krueger, 2016*) and were then mapped to human hg19 genome using Bowtie2 (*Langmead and Salzberg, 2012*). Germline variant calling was performed using GATK's HaplotypeCaller following the best practice guidelines (*McKenna et al., 2010*). Somatic variants were called using Strelka2 with a variant allele fraction (VAF) cut-off of 0.2 (35).

## DigiPico sequencing

DigiPico library preparation workflow is composed of five simple reaction steps performed in 384-well plate format (*Figure 1—figure supplement 2*). Addition of reagents can be performed using readily available liquid handling robots such as Mosquito HTS liquid handler (SPT Labtech, United Kingdom) (*Video 1*). For library preparation, 200 pg of purified DNA, 20–30 resuspended nuclei, or laser-capture micro-dissected tumor islets, were first denatured using 5 μl of D2 buffer from Repli-g single cell kit (Qiagen, USA). After 5 min incubation at room temperature, 95 μl of water was added to the sample and then 200 nl of the denatured template was added to each well of a 384-well

**Table 2.** Summary of patient samples and associated sequencing experiments.

| Patient ID | Sample type | Collection site | Analysis performed | |
| | | | Bulk WGS | DigiPico sequencing |
| --- | --- | --- | --- | --- |
| #11152 | Normal | Blood | x | |
| #11152 | Pre-chemo | Omentum | x | x |
| #11152 | Recurrence | Pelvic tumor | x | x |
| #11152 | Recurrence | Lymph node | | x |
| #11513 | Normal | Blood | x | x |
| #11513 | Pre-chemo | Ascites | x | x |
| OP1036 | Normal | Blood | x | |
| OP1036 | Pre-chemo | Paracolic gutter | x | x |
| #11502 | Normal | Blood | x | |
| #11502 | Pre-chemo | Peritoneum | x | x |

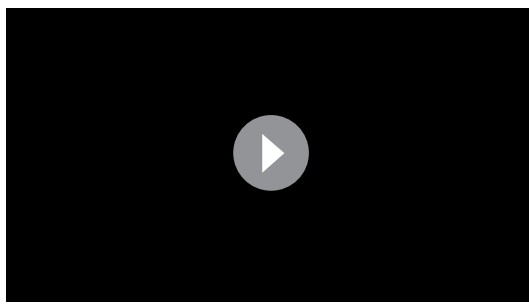

**Video 1.** Automated pipetting procedure of DigiPico library preparation.
https://elifesciences.org/articles/55207#video1

reaction plate already containing 800 nl of WGA mix (0.58 µl Sc Reaction Buffer, 0.04 µl Sc Polymerase (REPLI-g Single Cell kit, Qiagen), 0.075 µl 1 mM dUTP (Invitrogen, USA), 0.04 µl EvaGreen 20x (Biotium, USA), and 0.065 µl water). The plate was incubated at 30°C for 2 hr followed by heat inactivation at 65°C for 15 min. Addition of EvaGreen in the reaction allows for monitoring of the WGA reaction using a real-time PCR machine if required (*Hosokawa et al., 2017*). Next, controlled enzymatic fragmentation (*Peters et al., 2012*) reaction steps were sequentially performed on the whole genome amplified DNA without any purification steps. Briefly, (A) 1200 nl of UDG mix (0.08 U/µl rSAP (NEB, USA), 0.2 U/µl UDG (NEB, USA), 0.4 U/µl EndoIV (NEB, USA) in 1.8x NEBuffer 3) was added with 2 hr incubation at 37°C and heat inactivation at 65°C for 15 min. (B) 1200 nl of PolI mix (0.4 U/µl DNA Polymerase I (NEB, USA) 0.25 mM dNTP, 8 mM MgCl2, and 0.8 mM DTT) was added with 1.5 hr incubation at 37°C and heat inactivation at 70°C for 20 min. (C) 1200 nl of Klenow mix (0.5 U/µl Klenow exo- (NEB, USA), 0.5 mM dATP, 8 mM MgCl2, and 0.8 mM DTT) was added with 45 min incubation at 37°C and heat inactivation at 70°C for 20 min. (D) 400 nl of 20 µM full-length Illumina adapter oligos with well specific indices (*Figure 1—source data 4*) were added to each well followed by the addition of 1100 nl of Ligation mix (40 U/µl T4 DNA Ligase (NEB, USA), 5 mM ATP, 11.5% PEG 8000 (Qiagen, USA), and 6.8 mM MgCl2) and with 30 min incubation at 20°C and heat inactivation at 65°C for 15 min.

The resulting products were then pooled and the DNA was precipitated using an equal volume of isopropanol. DNA was then resuspended in water and the products were dual-size selected using Agencourt AMPure XP SPRI magnetic beads (Beckmann coulter) with 0.45x bead ratio for the left selection and an additional 0.32x for the right selection. The purified DNA was then resuspended in water and was immediately used for limited-cycle PCR amplification using the P5 and P7 primer mix (*Figure 1—source data 4*). PCR was performed for 12 cycles with 10 s annealing at 55°C and 45 s extension at 72°C. Final products were bead purified at 0.9x ratio. The resulting libraries were then sequenced on Illumina sequencing platforms in 2 × 150 paired-end sequencing mode to achieve a depth of coverage of 30 – 40x over human genome. The additional processing steps required for the DigiPico library preparation, at present, adds nearly £250 to the total reagent costs.

## Analysis of DigiPico sequencing data

The analysis pipeline of DigiPico sequencing data is presented in *Figure 1—figure supplement 3*. Briefly, 384 paired-read FastQ files for each well were obtained after demultiplexing the Illumina sequencing data. FastQ files were trimmed for adapter sequences and quality (*Krueger, 2016*). The first 12 nucleotides of each read were also removed. For mapping, in this work, we used Bowtie2 (*Langmead and Salzberg, 2012*) with the ignore-quals parameter activated mainly due to its higher speed. The ignore-quals parameter decreases the likelihood that low quality reads map to wrong locations in the genome. Reads were mapped to human hg19 reference genome and duplicate reads were marked using Picard Tools (*Picard Tools, 2018*). Joint variant calling was then performed on all 384 individual bam files together with a merged bam file from all wells using Platypus variant caller (*Rimmer et al., 2014*). The Platypus variant caller was chosen as it is currently the only algorithm capable of swiftly handling simultaneous callings on a large number of inputs which is essential for the analysis of DigiPico data. Next, all low-quality variants were removed by applying the quality filters (QUAL >60, FR >0.1, HP $\leq$ 4, QD >10, and SbPval $\leq$ 0.95). Moreover, the total number of wells covering each locus (Tw) and the number of wells supporting each variant (Vw) were determined and the well count filters (Tw >5, Vw > 2, and Vw/Tw >0.1) were applied to only retain the high confidence loci for analysis. Lastly, all regions of the genome with bad mappability (*Derrien et al., 2012*) were removed from the analysis using VCFtools (*Danecek et al., 2011*). The resulting list of high confidence de novo DigiPico variants were then used to perform variant re-calling (genotyping) on WGS data from blood and bulk of the tumor using Platypus with minPosterior

parameter set to 0 and minMapQual parameter set to 5. Any variant that was confidently unsupported in both of the standard WGS data, based on Platypus's joint variant calling results, was extracted as an UTD variant. Any variant that was confidently also present in the bulk sequencing data of the blood sample (based on GATK analysis) was extracted as a TP (True Positive) variant (*Figure 1—figure supplement 3*). Since only very high confidence germline variant calls based on bulk sequencing data are used in the labelling of the data at this stage, this step is impervious to the choice of germline variant caller.

## MutLX algorithm

The MutLX analysis pipeline is summarized in *Figure 2—figure supplement 1*.

### Artificial neural network architecture

The neural network model used in this study is a multilayer perceptron with an input layer consisting of N neurons (N = 41) where N is the number of features used in each experiment and was implemented in Python3 using Keras (*Chollet, 2015*). The model has two hidden layers with ReLU activations. We varied these numbers but did not see any significant improvement when using larger numbers of neurons. The last layer is a single output neuron with a sigmoid activation. The loss function is binary cross-entropy. For training, we applied a stochastic gradient descent optimization with momentum (Adam *Kingma and Ba, 2014*) with a learning rate of 0.001, a batch-size of 8 and for 10 epochs. After 10 epochs we did not observe any additional improvement in performance.

### Features used for training

The following features, extracted from the Platypus output of the DigiPico data, were used as the input of neural network model:

Platypus quality parameters: QUAL, BRF, FR, HP, HapScore, MGOF, MMLQ, MQ, QD, SbPval, NF, NR, TCF, and TCR (*Rimmer et al., 2014*).

Sequence context complexity: $F_{20}[1]$, $F_{20}[2]$, $F_{20}[3]$. Where $F_{20}[i]$ is the sum of the frequency of the $i$ most abundant nucleotides in the 10 bp sequence on either side of the variant position.

Read distribution data:

$R_{merge}[ref+var]$, and $R_{merge}[var]$ (where $R_{merge}[x]$ indicates the total number of reads in the merged bam file supporting the allele $x$. ref = reference allele; var = variant allele).

**W[ref >0 and var = 0][]**, **W[ref >0][0/0]**, **W[ref >0][0/1]**, **W[var >0][]**, **W[var >0][1/1]**, **W[var >0][0/1]**, **W[var >0 and ref = 0][0/1]**, and **W[ref >0 and var >0][]** (where **W[$i$][$j$]** is the number of wells matching criteria $i$ with genotype $j$. Absence of value for $j$ indicates all genotypes were considered).

W[ref = 0 and var >$n$][] (5 $\geq n \geq$ 0).

$R_{max}[m][var]$, and $R_{max}[m][ref+var]$ (3 $\geq n \geq$ 0; Where $R_{max}[y][x]$ shows the number of reads in the well with the $y^{th}$ highest number of reads supporting allele $x$).

$Max_c + Max_r$, and **W[var >0][] - ($Max_c$ + $Max_r$)** (where $Max_c$ is the number of variant supporting wells in the column with the highest number of wells supporting the variant allele and $Max_r$ is the number of variant supporting wells in the row with the highest number of wells supporting the variant allele).

### Training using MutLX

For each DigiPico run, we consider a full training set as the collection of all UTD variants (labelled as 0) and heterozygous germline SNPs (labelled as 1). The number of UTD variants in this set is much smaller than heterozygous germline SNPs, making the set imbalanced. Therefore, in order to avoid bias towards a specific label in the training we create 25 different balanced training subsets for each DigiPico run. This is done so that each training subset is composed of all UTD variants and a randomly selected subset of heterozygous germline SNPs with a size equal to the number of UTD variants. As explained previously, the majority of UTD variants are FP variant calls with an unknown ratio of true clone-specific variants among them, hence making the 0 labels noisy. To perform two-step training considering these noisy labels, we employ the following strategy. After training an initial model on each balanced training subset, the resulting model is applied to the mutations in the full training set to obtain an initial probability value for each mutation. These probability values indicate the predicted probability of a mutation belonging to label one category. Hence, any 0 labelled

mutation that attains a predicted probability value close to 1, is likely to be a mislabeled mutation. Therefore, to reduce the level of mislabeled data in the training set, all UTD variants with a probability value of more than 0.7 and all germline SNPs with a probability value of less than 0.3 are considered mislabeled and are removed from the training set. The cut-off values in this step were empirically determined by the analysis of various simulated datasets. In the end, following a similar sub-sampling strategy as in the initial training, a new model is trained on the remaining mutations of the training set. This model is then used for the analysis of all UTD variants.

## Calculation of probability and uncertainty scores

As explained earlier, in MutLX the training process is repeated 25 times with different randomly selected germline SNP subsets, resulting in different models each time and hence 25 different predicted probability values for each mutation. We therefore defined the 'probability score' for each mutation as the average of all of its predicted probability values:

$$\text{Probability score} = \frac{\sum_{i=1}^{n} P_i}{n}$$

Where $P_i$ is the probability value obtained from the $i^{th}$ training subset and n represents the number of subsets (n = 25).

Moreover, to obtain an uncertainty estimation for each probability value we performed a test-time drop-out analysis (*Gal and Ghahramani, 2015*). The trained model was applied to each mutation for 100 iterations during which, different neurons were dropped out with a rate of 0.8 and 0.7 for the first and the second hidden layers of the neural network, respectively. This process resulted in 100 probability values for each mutation. Based on these values, we defined the 'uncertainty score' for each mutation as the average of the dropout variances from the 25 different subsets:

$$\text{Uncertainty score} = \frac{\sum_{i=1}^{n} \sigma_i^2}{n}$$

Where $\sigma_i^2$ is the variance of 100 probability values obtained from the dropout analysis of the $i^{th}$ training subset and n represents the number of subsets (n = 25).

The uncertainty scores of all variants with a probability score above 0.2 (*Figure 2—figure supplement 4*) was used to generate a putative receiver operating characteristic (ROC) curve. The curve was generated by considering a range of cut-off values between 0.0 and 0.25 for the uncertainty score. At each cut-off value the ratio of germline SNPs that have an uncertainty score below the cut-off value was plotted against the corresponding number of UTDs. The area under the curve (AUC) was then calculated after normalizing the number of UTDs between 0 and 1. Note that in cases where true clone-specific variants are not expected (all UTDs are FP calls), this plot represents a ROC curve and the AUC of this plot should be close to 1, assuming a perfect model. In contrast, in when the AUC is significantly lower, it indicates the presence of true clone-specific variants in the sample. This negative correlation between the number of true UTDs and AUC was validated using simulated datasets (*Figure 2—figure supplement 5*). Based on these observations, for samples where the ROC curve suggests the presence of true clone-specific variants (AUC <0.9) MutLX uses 'uncertainty score' cut-off values that result in a TPR of 95% to improve the recovery rate of clone-specific variants. For datasets with an AUC ≥0.9 the cut-off value for filtering the data was determined based on the intersection of the threshold curve and the ROC curve.

## Generation and analysis of simulated DigiPico datasets

Simulated data were used to: (a) validate the negative correlation between the number of true UTDs and AUC (*Figure 2—figure supplement 5*) and (b) ensure that over-fitting to potentially true clone-specific variants does not occur (*Figure 2—figure supplement 2*).

To generate simulated datasets, we first identified somatic mutations in the bulk WGS data of the tumor sample PT2R from patient #11152 using Strelka2 somatic variant caller (*Kim et al., 2018*). These somatic variants were then identified in the de novo variant calling data of run D1110 and any somatic variant with a Tw > 6 and Vw/Tw > 0.45 was selected as a high-confidence somatic variant.

Next, various numbers of randomly selected high-confidence somatic variants were artificially mislabeled as UTDs (UTD*) to achieve 0.01, 0.02, 0.03, 0.04, 0.05, 0.06, 0.07, 0.08, 0.09, and 0.1 UTD*/UTD ratios. The resulting synthetic list of variants were then independently used for MutLX analysis and the number of UTDs and UTD*s that passed the MutLX filtering were calculated for each run. To ensure a robust analysis, for each ratio 10 different subsets of the somatic variants were analyzed. A similar analysis was also performed on the DigiPico data DE111, obtained from the bulk DNA extraction from an ascites sample of patient #11513.

## Validation of MutLX algorithm

Tumor sample PT2R from patient #11152 was used for the validation of the MutLX algorithm. A small piece of the tumor was macro-dissected from a frozen specimen and was embedded in OCT medium for sectioning. The first section (15 μm) from the tumor was collected in a separate tube and the nuclei were resuspended in 50 μl of sterile PBS solution. Total number of nuclei in the suspension was measured and a volume containing 30 nuclei was used for direct denaturation with an equal volume of D2 buffer from Repli-g mini WGA kit (Qiagen). The resulting crude lysate was directly used for DigiPico library preparation for run D1111. The remainder of the tumor sample was then used for bulk DNA extraction using DNeasy blood and tissue kit (Qiagen). 200 pg of the resulting DNA was directly used to prepare DigiPico library D1110. 1 μg of the DNA was used for standard library preparation using NEBNext Ultra DNA library preparation kit (NEB). In this setting only run D1111 is expected to have true clone-specific variants. Since the template for run D1110 is a subset of the template used in the bulk WGS analysis, nearly all real variants in run D1110 will also be present in the WGS data at similar frequencies and therefore will not be identified as UTDs. A similar logic is also applicable to the results of DigiPico runs DE011 and GM12885. Since both of these DigiPico runs had been performed on 200 pg of DNA from bulk DNA extractions, no true UTD variants are expected to be present in these samples. It is also worth noting that because of the digitized nature of the data, variants with very low frequencies (<0.05%) will show an inflated VAF in runs D1110, DE011, and GM12885, however, since such variants are unlikely to appear in more than one well, they will be eliminated from the data based on the Vw filter. Therefore, it is safe to assume that nearly all UTD variants in these runs are FP calls.

## Application of SCcaller on DigiPico data

SCcaller has originally been developed for the analysis of multiple displacement amplified single cell sequencing data (*Dong et al., 2017*). Since DigiPico library preparation also requires multiple displacement amplification on limiting amounts of template DNA, the resulting data are fundamentally similar to the natural input for SCcaller. Therefore, we used the merged bam file of DigiPico data as an input for SCcaller. For the analysis, the list of heterozygous SNPs was obtained from the respective bulk WGS data using GATK HaplotypeCaller and cut-off values were used for alpha = 0.01. Next, all filtered SNVs were used for variant re-calling on respective standard WGS data and all variants that were confidently unsupported by WGS data were extracted as UTD variants.

## Mutation validation

Variants that pass the MutLX analysis were validated by comparison with deep sequencing data of the bulk tumor from an independent sequencing platform. All DigiPico data from patient #11152 were validated through comparison with 39 deep sequencing datasets obtained from the same tumor masses sequenced on Complete Genomics sequencing platform (*Drmanac et al., 2010*). This included three Complete Genomics bulk sequencing and 36 LFR (Long-Fragment Read) sequencing data. Since the independent sequencing data for the omental mass were not obtained from exactly the same tumor mass as the ones that were used for DigiPico sequencing the validation rate by such a comparison for these runs is not expected to be high. For targeted validation, primers were designed to obtain amplicons containing the variants using the primer3 tool (*Figure 2—source data 5*). Amplicons were obtained by performing a 2-step PCR using Phusion High-Fidelity PCR Master Mix with GC Buffer for 16 cycles on 1 ng of template. All amplicons from each sample were then pooled and purified before adapter ligation and indexing using NEBNext Ultra II kit. The resulting libraries were sequenced on a MiSeq platform. Sequencing results were mapped to human hg19

genome using Bowtie2 and the number of reads supporting each variant was counted using Platypus variant caller.

## Local hyper-mutation (kataegis) analysis

To generate the rainfall plots, the distances between pairs of consecutive somatic mutations on chromosome 17 were plotted against their genomic position of the second mutation in each pair using a custom script in R. The presence of clusters of localized mutations indicates kataegis events. In these plots, each dot is colored based on the mutation type of the second mutation in the pair in respect to the hg19 human reference genome.

## Code availability

Source code for MutLX is available on Github at https://github.com/mmdknr/DigiPico (*Carrami and Sharifzadeh, 2020*; copy archived at https://github.com/elifesciences-publications/DigiPico).

## Accession numbers

Sequence data has been deposited at the European Genome-phenome Archive (EGA), which is hosted by the EBI and the CRG, under accession number EGAS00001003555 (EGAD00001005118). Further information about EGA can be found on https://ega-archive.org 'The European Genome-phenome Archive of human data consented for biomedical research'.

## Acknowledgements

We thank Professor Robert C Bast Jr. and Professor Vincenzo Cerundolo for helpful discussions. We thank Dr Neil Ashley for technical support in using the Mosquito liquid handler. We are grateful to Dr Donatien C Fotso and Ewan Mac Mahon for bioinformatics support. We thank Dr Sunanda Dhar for support with histopathology.

## Additional information

### Competing interests

Eli M Carrami: AA and EMC hold a patent application for DigiPico sequencing method (UK Patent Application No. 1918043.9). Ahmed A Ahmed: EMC and AA have filed a patent application regarding the DigiPico method (UK Patent Application No. 1918043.9). The other authors declare that no competing interests exist.

### Funding

| Funder | Grant reference number | Author |
| --- | --- | --- |
| Ovarian Cancer Action | HER000762 | Ahmed A Ahmed |
| National Institute for Health Research | IS-BRC-0211-10025 | Ahmed A Ahmed |
| Helen Clarke Fund | HENBZPE1 | Ahmed A Ahmed |

The funders had no role in study design, data collection and interpretation, or the decision to submit the work for publication.

### Author contributions

Eli M Carrami, Conceptualization, Data curation, Software, Formal analysis, Validation, Visualization, Methodology, Writing - original draft, Writing - review and editing; Sahand Sharifzadeh, Conceptualization, Software, Methodology, Writing - review and editing; Nina C Wietek, Resources, Visualization, Writing - review and editing; Mara Artibani, Salma El-Sahhar, Writing - review and editing; Tatjana Sauka-Spengler, Supervision; Christopher Yau, Resources, Funding acquisition; Volker Tresp, Conceptualization, Supervision, Writing - review and editing; Ahmed A Ahmed, Conceptualization,

Resources, Data curation, Formal analysis, Supervision, Funding acquisition, Visualization, Project administration, Writing - review and editing

### Author ORCIDs
Eli M Carrami  https://orcid.org/0000-0002-7770-2065
Tatjana Sauka-Spengler  http://orcid.org/0000-0001-9289-0263
Christopher Yau  http://orcid.org/0000-0001-7615-8523
Ahmed A Ahmed  https://orcid.org/0000-0001-6509-2581

### Ethics

Human subjects: Patients #11152, #11502 and #11513 provided written consent for participation in the prospective biomarker validation study Gynaecological Oncology Targeted Therapy Study 01 (GO-Target-01) under research ethics approval number 11/SC/0014. Patient OP1036 participated in the prospective Oxford Ovarian Cancer Predict Chemotherapy Response Trial (OXO-PCR-01), under research ethics approval number 12/SC/0404. Necessary informed consents from study participants were obtained as appropriate.

### Decision letter and Author response
Decision letter https://doi.org/10.7554/eLife.55207.sa1
Author response https://doi.org/10.7554/eLife.55207.sa2

## Additional files

### Supplementary files
• Transparent reporting form

### Data availability

Sequence data has been deposited at the European Genome-phenome Archive (EGA), which is hosted by the EBI and the CRG, under accession number EGAS00001003555 (EGAD00001005118). Further information about EGA can be found on https://ega-archive.org "The European Genome-phenome Archive of human data consented for biomedical research".

The following dataset was generated:

| Author(s) | Year | Dataset title | Dataset URL | Database and Identifier |
|---|---|---|---|---|
| Carrami EM, Ahmed AA | 2019 | DigiPico sequencing data for the study of active mutational processes in HGSOC | https://ega-archive.org/datasets/EGAD00001005118 | European Genome-phenome Archive, EGAD00001005118 |

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
