## [Decision Letter]

**Acceptance summary:**

We and the reviewers were impressed and excited about the possibilities that DigiPico/MutLX offers for studying active mutational processes in tumours and believe that it will be quickly taken up by the scientific community. Regarding your manuscript, we found it very clear and easy to follow, and the figures quite pleasing and effective at communicating your method, your ideas and your results. So we just want to add, congratulations on a great paper!

**Decision letter after peer review:**

Thank you for submitting your work entitled "Revealing active mutational processes in tumours using DigiPico/MutLX at unprecedented accuracy" for consideration by *eLife*. Your article has been reviewed by three peer reviewers, one of whom is a member of our Board of Reviewing Editors, and the evaluation has been overseen by Maureen Murphy (Senior Editor).

The manuscript has been favourably evaluated by the reviewers but there are some issues that need to be addressed before acceptance, as outlined below. Please give your most serious and rigorous attention to these comments, as many of them speak to the ability for others to follow and utilize your work, and this is a critical issue.

Major points:

1) Please discuss the following question. Can the algorithm be used for publicly available single cell WGS datasets if the sequencing depth is high enough or is the sample preparation absolutely required?

2) Please give an indication of how difficult it would be to implement this technique, especially the library preparation step – are people with special training required?

3) Please add a point in the Discussion section regarding previous approaches: neither the sequencing nor machine learning approach are discussed in the comprehensive context of previous developments. For example: Is this the first application of ANNs to detect true somatic mutations? Possibly not. How is this approach then different from previous ones? Please also discuss other similar workflows that may have been developed regarding the sequencing method.

4) The reviewers have agreed that the title of the manuscript does not reflect its contents accurately. The current title seems to promise an in-depth study into active mutational processes, while the focus of the paper is actually on the technology. The reviewers suggest the title "A combined digital sequencing and neural network approach identifies mutations in single cells with high accuracy" would be much closer to the paper’s contributions. In addition, the manuscript needs a discussion of what is known in the mutational processes field in ovarian cancer and how the DigiPico method complements existing copy number-based methods, and what a better characterisation of SNVs will add to our understanding of this copy number disease.

---

## [Author Response]

Major points:1) Please discuss the following question. Can the algorithm be used for publicly available single cell WGS datasets if the sequencing depth is high enough or is the sample preparation absolutely required?

This point has now been clarified in the Discussion section as follows:

“MutLX takes advantage of compartment level data generated through the DigiPico library preparation method followed by, relatively, deep sequencing of each compartment. However, in theory, other linked-read library preparation methods such as the commercially available 10x platform may also generate data that could be utilized by MutLX, if modifications are applied. For such alternative approaches to benefit from MutLX the DNA input should be lowered, and the depth of sequencing per each compartment must significantly increase.”

2) Please give an indication of how difficult it would be to implement this technique, especially the library preparation step – are people with special training required?

The workflow for DigiPico library preparation method is very straightforward and can be easily implemented on any automated liquid handling instrument. In our research, we have used the Mosquito liquid handler. The method can be performed by anyone and does not require specialist training. To illustrate the simplicity of the method we have now added a supplementary figure showing the library preparation workflow as well as a video to show the simplicity of implementing the method using liquid handling instruments such as Mosquito in our case. Also, to emphasize this point, the following statement was added:

“DigiPico library preparation workflow is composed of five simple reaction steps performed in 384-well plate format (Figure 1—figure supplement 2). Addition of reagents can be performed using readily available liquid handling robots such as Mosquito HTS liquid handler (TTP Labtech, Australia) (Video 1).”

The analysis pipeline for DigiPico data only requires basic NGS analysis tools, as stated in the Materials and methods section. The MutLX analysis algorithm is available as open-source code on Github with example and test files. The entire MutLX analysis pipeline can be launched on the input data using a single command line.

3) Please add a point in the Discussion section regarding previous approaches: neither the sequencing nor machine learning approach are discussed in the comprehensive context of previous developments. For example: Is this the first application of ANNs to detect true somatic mutations? Possibly not. How is this approach then different from previous ones? Please also discuss other similar workflows that may have been developed regarding the sequencing method.

To address this comment the following text and relevant references were added in the manuscript:

“While ANNs and other machine learning approaches have frequently been employed in the field of genomics, the focus of these methodologies has mainly been on the elimination of technical and computational noise from sequencing data (23-27). Such sources of noise, resulting from low-quality sequencing data or inaccurate mapping of short reads, lead to variant calls that had not been present in the sequenced library and their elimination is essential for accurate identification of somatic mutations. However, these methods are unable to address the issue of artefactual mutations that physically occur in the sequenced libraries due to the WGA of damaged DNA. As such, artefactual mutations will exhibit all the features of real mutations and will not be eliminated by the previously described algorithms. Given that the process of WGA "fixates" such mutations into the final library, it is essential to preserve some information prior to the start of the WGA which can later be employed for identification of these sources of error. In our method, we preserve this information by compartmentalizing the template DNA prior to the start of WGA and by subsequent indexing of the compartments during the DigiPico library preparation pipeline. As a result, in MutLX rather than trying to evaluate whether a variant call had indeed been present in the library, we use the complex pre-WGA information to decipher whether the called mutations had existed in the original template or they have artefactually arisen through the process of WGA. This critical distinction is what allows us to identify extremely low-prevalence mutations in small populations of cells which in turn enables the study of active mutational processes.”

4) The reviewers have agreed that the title of the manuscript does not reflect its contents accurately. The current title seems to promise an in-depth study into active mutational processes, while the focus of the paper is actually on the technology. The reviewers suggest the title "A combined digital sequencing and neural network approach identifies mutations in single cells with high accuracy" would be much closer to the paper’s contributions. In addition, the manuscript needs a discussion of what is known in the mutational processes field in ovarian cancer and how the DigiPico method complements existing copy number-based methods, and what a better characterisation of SNVs will add to our understanding of this copy number disease.

We thank the reviewers for this suggestion. We want to suggest the following title “A highly accurate platform for clone-specific mutation discovery enables the study of active mutational processes”. We believe this title is appropriate since it reflects your suggestion as well as indicating that DigiPico is an enabling technology for the study of active mutational processes without promising an in-depth study. Please consider that we do provide proof of concept data (see Figure 3) to show that this approach can, in fact, enable the investigation of active mutational processes.